# $\Sigma$-Optimality for Active Learning on Gaussian Random Fields

**Yifei Ma**
Machine Learning Department
Carnegie Mellon University
yifeim@cs.cmu.edu

**Roman Garnett**
Computer Science Department
University of Bonn
rgarnett@uni-bonn.de

**Jeff Schneider**
Robotics Institute
Carnegie Mellon University
schneide@cs.cmu.edu

## Abstract

A common classifier for unlabeled nodes on undirected graphs uses label propagation from the labeled nodes, equivalent to the harmonic predictor on Gaussian random fields (GRFs). For active learning on GRFs, the commonly used V-optimality criterion queries nodes that reduce the $L^2$ (regression) loss. V-optimality satisfies a submodularity property showing that greedy reduction produces a $(1-1/e)$ globally optimal solution. However, $L^2$ loss may not characterise the true nature of 0/1 loss in classification problems and thus may not be the best choice for active learning.

We consider a new criterion we call $\Sigma$-optimality, which queries the node that minimizes the sum of the elements in the predictive covariance. $\Sigma$-optimality directly optimizes the risk of the surveying problem, which is to determine the proportion of nodes belonging to one class. In this paper we extend submodularity guarantees from V-optimality to $\Sigma$-optimality using properties specific to GRFs. We further show that GRFs satisfy the *suppressor-free condition* in addition to the conditional independence inherited from Markov random fields. We test $\Sigma$-optimality on real-world graphs with both synthetic and real data and show that it outperforms V-optimality and other related methods on classification.

## 1 Introduction

Real-world data are often presented as a graph where the nodes in the graph bear labels that vary smoothly along edges. For example, for scientific publications, the content of one paper is highly correlated with the content of papers that it references or is referenced by, the field of interest of a scholar is highly correlated with other scholars s/he coauthors with, etc. Many of these networks can be described using an undirected graph with nonnegative edge weights set to be the strengths of the connections between nodes.

The model for label prediction in this paper is the harmonic function on the Gaussian random field (GRF) by Zhu et al. (2003). It can generalize two popular and intuitive algorithms: label propagation (Zhu & Ghahramani, 2002), and random walk with absorptions (Wu et al., 2012). GRFs can be seen as a Gaussian process (GP) (Rasmussen & Williams, 2006) with its (maybe improper) prior covariance matrix whose (pseudo)inverse is set to be the graph Laplacian.

Like other learning problems, labels may be insufficient and expensive to gather, especially if one wants to discover a new phenomenon on a graph. Active learning addresses these issues by making automated decisions on which nodes to query for labels from experts or the crowd. Some popular criteria are empirical risk minimization (Settles, 2010; Zhu et al., 2003), mutual information gain (Krause et al., 2008), and V-optimality (Ji & Han, 2012). Here we consider an alternative criterion, $\Sigma$-optimality, and establish several related theoretical results. Namely, we show that greedy reduction of $\Sigma$-optimality provides a $(1 - 1/e)$ approximation bound to the global optimum. We also show

that Gaussian random fields satisfy the suppressor-free condition, described below. Finally, we show that $\Sigma$-optimality outperforms other approaches for active learning with GRFs for classification.

## 1.1 V-optimality on Gaussian Random Fields

Ji & Han (2012) proposed greedy variance minimization as a cheap and high profile surrogate active classification criterion. To decide which node to query next, the active learning algorithm finds the unlabeled node which leads to the smallest average predictive variance on all other unlabeled nodes. It corresponds to standard V-optimality in optimal experiment design.

We will discuss several aspects of V-optimality on GRFs below: 1. The motivation behind V-optimality can be paraphrased as the expected risk minimization with the $L^2$-surrogate loss (Section 2.1). 2. The greedy solution to the set optimization problem in V-optimality is comparable to the global solution up to a constant (Theorem 1). 3. The greedy application of V-optimality can also be interpreted as a heuristic which selects nodes that have high correlation to nodes with high variances (Observation 4).

Some previous work is related to point 2 above. Nemhauser et al. (1978) shows that any *submodular*, monotone and normalized set function yields a $(1 - 1/e)$ global optimality guarantee for greedy solutions. Our proof techniques coincides with Friedland & Gaubert (2011) in principle, but we are not restricted to spectral functions. Krause et al. (2008) showed a counter example where the V-optimality objective function with GP models does not satisfy submodularity.

## 1.2 $\Sigma$-optimality on Gaussian Random Fields

We define $\Sigma$-optimality on GRFs to be another variance minimization criterion that minimizes the sum of all entries in the predictive covariance matrix. As we will show in Lemma 7, the predictive covariance matrix is nonnegative entry-wise and thus the definition is proper. $\Sigma$-optimality was originally proposed by Garnett et al. (2012) in the context of *active surveying*, which is to determine the proportion of nodes belonging to one class. However, we focus on its performance as a criterion in active classification heuristics. The survey-risk of $\Sigma$-optimality replaces the $L^2$-risk of V-optimality as an alternative surrogate risk for the 0/1-risk.

We also prove that the greedy application of $\Sigma$-optimality has a similar theoretical bound as V-optimality. We will show that greedily minimizing $\Sigma$-optimality empirically outperforms greedily minimizing V-optimality on classification problems. The exact reason explaining the superiority of $\Sigma$-optimality as a surrogate loss in the GRF model is still an open question, but we observe that $\Sigma$-optimality tends to select cluster centers whereas V-optimality goes after outliers (Section 3.1). Finally, greedy application of both $\Sigma$-optimality and V-optimality need $\mathcal{O}(N)$ time per query candidate evaluation after one-time inverse of a $N \times N$ matrix.

## 1.3 GRFs Are Suppressor Free

In linear regression, an explanatory variable is called a suppressor if adding it as a new variable enhances correlations between the old variables and the dependent variable (Walker, 2003; Das & Kempe, 2008). Suppressors are persistent in real-world data. We show GRFs to be *suppressor-free*. Intuitively, this means that with more labels acquired, the conditional correlation between unlabeled nodes decreases even when their Markov blanket has not formed. That GRFs present natural examples for the otherwise obscure suppressor-free condition is interesting.

## 2 Learning Model & Active Learning Objectives

We use *Gaussian random field/label propagation* (GRF/LP) as our learning model. Suppose the dataset can be represented in the form of a connected undirected graph $G = (V, E)$ where each node has an (either known or unknown) label and each edge $e_{ij}$ has a fixed nonnegative weight $w_{ij}(= w_{ji})$ that reflects the proximity, similarity, etc. between nodes $v_i$ and $v_j$. Define the graph Laplacian of $G$ to be $L = \text{diag}(W\mathbf{1}) - W$, i.e., $l_{ii} = \sum_j w_{ij}$ and $l_{ij} = -w_{ij}$ when $i \neq j$. Let $L_\delta = L + \delta I$ be the generalized Laplacian obtained by adding self-loops. In the following, we will write $L$ to also encompass $\beta L_\delta$ for the set of hyper-parameters $\beta > 0$ and $\delta \geq 0$.

The *binary* GRF is a Bayesian model to generate $y_i \in \{0, +1\}$ for every node $v_i$ according to,

$$p(\boldsymbol{y}) \propto \exp\left\{ -\frac{\beta}{2}\Big(\sum_{i,j} w_{ij}(y_i - y_j)^2 + \delta \sum_i y_i^2\Big)\right\} = \exp\left(-\frac{1}{2}\boldsymbol{y}^T L \boldsymbol{y}\right). \qquad (2.1)$$

Suppose nodes $\boldsymbol{\ell} = \{v_{\ell_1}, \ldots, v_{\ell_{|\boldsymbol{\ell}|}}\}$ are labeled as $\boldsymbol{y_\ell} = (y_{\ell_1}, \ldots, y_{\ell_{|\ell|}})^T$; A GRF infers the output distribution on unlabeled nodes, $\boldsymbol{y_u} = (y_{u_1}, \ldots, y_{u_{|u|}})^T$ by the conditional distribution given $\boldsymbol{y_\ell}$, as

$$\Pr(\boldsymbol{y_u}|\boldsymbol{y_\ell}) \propto \mathcal{N}(\hat{\boldsymbol{y}}_{\boldsymbol{u}}, L_{\boldsymbol{u}}^{-1}) = \mathcal{N}(\hat{\boldsymbol{y}}_{\boldsymbol{u}}, L_{(\boldsymbol{v}-\boldsymbol{\ell})}^{-1}), \qquad (2.2)$$

where $\hat{\boldsymbol{y}}_{\boldsymbol{u}} = (-L_{\boldsymbol{u}}^{-1} L_{\boldsymbol{u}\boldsymbol{\ell}} \boldsymbol{y_\ell})$ is the vector of predictive means on unlabeled nodes and $L_{\boldsymbol{u}}$ is the principal submatrix consisting of the unlabeled row and column indices in $L$, that is, the lower-right block of $L = \begin{pmatrix} L_{\boldsymbol{\ell}} & L_{\boldsymbol{\ell}\boldsymbol{u}} \\ L_{\boldsymbol{u}\boldsymbol{\ell}} & L_{\boldsymbol{u}} \end{pmatrix}$. By convention, $L_{(\boldsymbol{v}-\boldsymbol{\ell})}^{-1}$ means the inverse of the principal submatrix. We use $L_{(\boldsymbol{v}-\boldsymbol{\ell})}$ and $L_{\boldsymbol{u}}$ interchangeably because $\boldsymbol{\ell}$ and $\boldsymbol{u}$ partition the set of all nodes $\boldsymbol{v}$.

Finally, GRF, or GRF/LP, is a relaxation of the *binary* GRF to continuous outputs, because the latter is computationally intractable even for *a-priori* generations. LP stands for label propagation, because the predictive mean on a node is the probability of a random walk leaving that node hitting a positive label before hitting a zero label. For multi-class problems, Zhu et al. (2003) proposed the *harmonic predictor* which looks at predictive means in one-versus-all comparisons.

**Remark:** An alternative approximation to the *binary* GRF is the GRF-sigmoid model, which draws the binary outputs from Bernoulli distributions with means set to be the sigmoid function of the GRF (latent) variables. However, this alternative is very slow to compute and may not be compatible with the theoretical results in this paper.

## 2.1 Active Learning Objective 1: $L^2$ Risk Minimization (V-Optimality)

Since in GRFs, regression responses are taken directly as probability predictions, it is computationally and analytically more convenient to apply the regression loss directly in the GRF as in Ji & Han (2012). Assume the $L^2$ loss to be our classification loss. The risk function, whose input variable is the labeled subset $\boldsymbol{\ell}$, is:

$$R_V(\boldsymbol{\ell}) = \mathbb{E}^{\boldsymbol{y_\ell y_u}} \sum_{u_i \in \boldsymbol{u}} (y_{u_i} - \hat{y}_{u_i})^2 = \mathbb{E}\left[\mathbb{E}\left[\sum_{u_i \in \boldsymbol{u}} (y_{u_i} - \hat{y}_{u_i})^2 \bigg| \boldsymbol{y_\ell}\right]\right] = \operatorname{tr}(L_{\boldsymbol{u}}^{-1}). \qquad (2.3)$$

This risk is written with a subscript $V$ because minimizing (2.3) is also the V-optimality criterion, which minimizes mean prediction variance in active learning.

In active learning, we strive to select a subset $\boldsymbol{\ell}$ of nodes to query for labels, constrained by a given budget $C$, such that the risk is minimized. Formally,

$$\underset{\boldsymbol{\ell}:\ |\boldsymbol{\ell}| \leq C}{\arg\min} \quad R(\boldsymbol{\ell}) = R_V(\boldsymbol{\ell}) = \operatorname{tr}(L_{(\boldsymbol{v}-\boldsymbol{\ell})}^{-1}). \qquad (2.4)$$

## 2.2 Active Learning Objective 2: Survey Risk Minimization ($\Sigma$-Optimality)

Another objective building on the GRF model (2.2) is to determine the proportion of nodes belonging to class 1, as would happen when performing a survey. For active surveying, the risk would be:

$$R_\Sigma(\boldsymbol{\ell}) = \mathbb{E}^{\boldsymbol{y_\ell y_u}}\Big(\sum_{u_i \in \boldsymbol{u}} y_{u_i} - \sum_{u_i \in \boldsymbol{u}} \hat{y}_{u_i}\Big)^2 = \mathbb{E}\big[\mathbb{E}\big[\big(\mathbf{1}^T \boldsymbol{y_u} - \mathbf{1}^T \hat{\boldsymbol{y}}_{\boldsymbol{u}}\big)^2 | \boldsymbol{y_\ell}\big]\big] = \mathbf{1}^T L_{\boldsymbol{u}}^{-1} \mathbf{1}, \qquad (2.5)$$

which could substitute the risk $R(\boldsymbol{\ell})$ in (2.4) and yield another heuristic for selecting nodes in batch active learning. We will refer to this modified optimization objective as the $\Sigma$-optimality heuristic:

$$\underset{\boldsymbol{\ell}:\ |\boldsymbol{\ell}| \leq C}{\arg\min} \quad R(\boldsymbol{\ell}) = R_\Sigma(\boldsymbol{\ell}) = \mathbf{1}^T L_{(\boldsymbol{v}-\boldsymbol{\ell})}^{-1} \mathbf{1}. \qquad (2.6)$$

Further, we will also consider the application of $\Sigma$-optimality in active classification because (2.6) is another metric of the predictive variance. Surprisingly, although both (2.3) and (2.5) are approximations of the real objective (the 0/1 risk), greedy reduction of the $\Sigma$-optimality criterion outperforms greedy reduction of the V-optimality criterion in active classification (Section 3.1 and 5.1), as well as several other methods including expected error reduction.

## 2.3 Greedy Sequential Application of V/Σ-Optimality

Both (2.4) and (2.6) are subset optimization problems. Calculating the global optimum may be intractable. As will be shown later in the theoretical results, the reduction of both risks are submodular set functions and the greedy sequential update algorithm yields a solution that has a guaranteed approximation ratio to the optimum (Theorem 1).

At the $k$-th query decision, denote the covariance matrix conditioned on the previous $(k-1)$ queries as $C = (L_{(\boldsymbol{v}-\boldsymbol{\ell}^{(k-1)})})^{-1}$. By Shur's Lemma (or the GP-regression update rule), the one-step look-ahead covariance matrix conditioned on $\boldsymbol{\ell}^{(k-1)} \cup \{v\}$, denoted as $C' = (L_{(\boldsymbol{v}-(\boldsymbol{\ell}^{(k-1)}\cup\{v\}))})^{-1}$, has the following update formula:

$$\begin{pmatrix} C' & 0 \\ 0 & 0 \end{pmatrix} = C - \frac{1}{C_{vv}} \cdot C_{:v} C_{v:}, \tag{2.7}$$

where without loss of generality $v$ was positioned as the last node. Further denoting $C_{ij} = \rho_{ij}\sigma_i\sigma_j$, we can put (2.7) inside $R_\Sigma(\cdot)$ and $R_V(\cdot)$ to get the following equivalent criteria:

$$\text{V-optimality} : \quad v_*^{(k)} = \arg\max_{v\in\boldsymbol{u}} \frac{\sum_{t\in\boldsymbol{u}}(C_{vt})^2}{C_{vv}} = \sum_{t\in\boldsymbol{u}} \rho_{vt}^2\sigma_t^2, \tag{2.8}$$

$$\Sigma\text{-optimality} : \quad v_*^{(k)} = \arg\max_{v\in\boldsymbol{u}} \frac{(\sum_{t\in\boldsymbol{u}} C_{vt})^2}{C_{vv}} = (\sum_{t\in\boldsymbol{u}} \rho_{vt}\sigma_t)^2. \tag{2.9}$$

# 3 Theoretical Results & Insights

For the general GP model, greedy optimization of the $L^2$ risk has no guarantee that the solution can be comparable to the brute-force global optimum (taking exponential time to compute), because the objective function, the trace of the predictive covariance matrix, fails to satisfy submodularity in all cases (Krause et al., 2008). However, in the special case of GPs with kernel matrix equal to the inverse of a graph Laplacian (with $\boldsymbol{\ell} \neq \emptyset$ or $\delta > 0$), the GRF does provide such theoretical guarantees, both for V-optimality and Σ-optimality. The latter is a novel result.

The following theoretical results concern greedy maximization of the risk reduction function (which is shown to be submodular): $R_\Delta(\boldsymbol{\ell}) = R(\emptyset) - R(\boldsymbol{\ell})$ for either $R(\cdot) = R_V(\cdot)$ or $R_\Sigma(\cdot)$.

**Theorem 1** (Near-optimal guarantee for greedy applications of V/Σ-optimality). *In risk reduction,*

$$R_\Delta(\boldsymbol{\ell}_g) \geq (1 - 1/e) \cdot R_\Delta(\boldsymbol{\ell}_*), \tag{3.1}$$

*where $R_\Delta(\boldsymbol{\ell}) = R(\emptyset) - R(\boldsymbol{\ell})$ for either $R(\cdot) = R_V(\cdot)$ or $R_\Sigma(\cdot)$, e is Euler's number, $\boldsymbol{\ell}_g$ is the greedy optimizer, and $\boldsymbol{\ell}_*$ is the true global optimizer under the constraint $|\boldsymbol{\ell}_*| \leq |\boldsymbol{\ell}_g|$.*

According to Nemhauser et al. (1978), it suffices to show the following properties of $R_\Delta(\boldsymbol{\ell})$:

**Lemma 2** (Normalization, Monotonicity, and Submodularity). $\forall \boldsymbol{\ell}_1 \subset \boldsymbol{\ell}_2 \subset \boldsymbol{v}, v \in \boldsymbol{v}$,

$$R_\Delta(\emptyset) = 0, \tag{3.2}$$

$$R_\Delta(\boldsymbol{\ell}_2) \geq R_\Delta(\boldsymbol{\ell}_1), \tag{3.3}$$

$$R_\Delta(\boldsymbol{\ell}_1 \cup \{v\}) - R_\Delta(\boldsymbol{\ell}_1) \geq R_\Delta(\boldsymbol{\ell}_2 \cup \{v\}) - R_\Delta(\boldsymbol{\ell}_2). \tag{3.4}$$

Another sufficient condition for Theorem 1, which is itself an interesting observation, is the *suppressor-free* condition. Walker (2003) describes a *suppressor* as a variable, knowing which will suddenly create a strong correlation between the predictors. An example is $y_i + y_j = y_k$. Knowing any one of these will create correlations between the others. Walker further states that suppressors are common in regression problems. Das & Kempe (2008) extend the suppressor-free condition to sets and showed that this condition is sufficient to prove (2.3). Formally, the condition is:

$$\big|\text{corr}(y_i, y_j \mid \boldsymbol{\ell}_1 \cup \boldsymbol{\ell}_2)\big| \leq \big|\text{corr}(y_i, y_j \mid \boldsymbol{\ell}_1)\big|$$
$$\forall v_i, v_j \in \boldsymbol{v}, \forall \boldsymbol{\ell}_1, \boldsymbol{\ell}_2 \subset \boldsymbol{v}. \tag{3.5}$$

It may be easier to understand (3.5) as a decreasing correlation property. It is well known for Markov random fields that the labels of two nodes on a graph become independent given labels of their Markov blanket. Here we establish that GRF boasts more than that: the correlation between any two nodes decreases as more nodes get labeled, even before a Markov blanket is formed. Formally:

**Theorem 3** (Suppressor-Free Condition)**.** (3.5) *holds for pairs of nodes in the* GRF *model. Note that since the conditional covariance of the* GRF *model is* $L_{(\boldsymbol{v}-\boldsymbol{\ell})}^{-1}$*, we can properly define the corresponding conditional correlation to be*

$$\text{corr}(\boldsymbol{y_u}|\boldsymbol{\ell}) = D^{-\frac{1}{2}} L_{(\boldsymbol{v}-\boldsymbol{\ell})}^{-1} D^{-\frac{1}{2}}, \text{ with } D = diag\left(L_{(\boldsymbol{v}-\boldsymbol{\ell})}^{-1}\right). \tag{3.6}$$

### 3.1 Insights From Comparing the Greedy Applications of the $\Sigma$/V-Optimality Criteria

Both the V/$\Sigma$-optimality are approximations to the 0/1 risk minimization objective. Unfortunately, we cannot theoretically reason why greedy $\Sigma$-optimality outperforms V-optimality in the experiments. However, we made two observations during our investigation that provide some insights. An illustrative toy example is also provided in Section 5.1.

**Observation 4.** *Eq.* (2.8) *and* (2.9) *suggest that both the greedy* $\Sigma$/V-*optimality selects nodes that (1) have high variance and (2) are highly correlated to high-variance nodes, conditioned on the labeled nodes. Notice Lemma 7 proves that predictive correlations are always nonnegative.*

In order to contrast $\Sigma$/V-optimality, rewrite (2.9) as:

$$(\Sigma\text{-optimality}) : \ \underset{v\in\boldsymbol{u}}{\arg\max}\ \left(\textstyle\sum_{t\in\boldsymbol{u}}\rho_{vt}\sigma_t\right)^2 = \textstyle\sum_{t\in\boldsymbol{u}}\rho_{vt}^2\sigma_t^2 + \sum_{t_1\neq t_2\in\boldsymbol{u}}\rho_{vt_1}\rho_{vt_2}\sigma_{t_1}\sigma_{t_2}. \tag{3.7}$$

**Observation 5.** $\Sigma$-*optimality has one more term that involves cross products of* $(\rho_{vt_1}\sigma_{t_1})$ *and* $(\rho_{vt_2}\sigma_{t_2})$ *(which are nonnegative according to Lemma 9). By the Cauchy–Schwartz Inequality, the sum of these cross products are maximized when they are equal. So, the* $\Sigma$-*optimality additionally favors nodes that (3) have consistent global influence, i.e., that are more likely to be in cluster centers.*

## 4 Proof Sketches

Our results predicate on and extend to GPs whose inverse covariance matrix meets Proposition 6.

**Proposition 6.** *L satisfies the following.* [1]

| # | Textual description | Mathematical expression |
|---|---|---|
| p6.1 | *L has proper signs.* | $l_{ij} \geq 0$ if $i = j$ and $l_{ij} \leq 0$ if $i \neq j$. |
| p6.2 | *L is undirected and connected.* | $l_{ij} = l_{ji}\forall i, j$ and $\sum_{j\neq i}(-l_{ij}) > 0$. |
| p6.3 | *Node degree no less than number of edges.* | $l_{ii} \geq \sum_{j\neq i}(-l_{ij}) = \sum_{j\neq i}(-l_{ji}) > 0, \forall i$. |
| p6.4 | *L is nonsingular and positive-definite.* | $\exists i : \ l_{ii} > \sum_{j\neq i}(-l_{ij}) = \sum_{j\neq i}(-l_{ji}) > 0$. |

Although the properties of V-optimality fall into the more general class of *spectral functions* (Friedland & Gaubert, 2011), we have seen no proof of either the suppressor-free condition or the submodularity of $\Sigma$-optimality on GRFs. We write the ideas behind the proofs. Details are in the appendix.[2]

**Lemma 7.** *For any L satisfying (p6.1-4),* $L^{-1} \geq 0$ *entry-wise.*[3]

*Proof.* Sketch: Suppose $L = D - W = D(I - D^{-1}W)$, with $D = \text{diag}(L)$. Then we can show the convergence of the Taylor expansion (Appendix A.1):

$$L^{-1} = [I + \textstyle\sum_{r=1}^{\infty}(D^{-1}W)^r]D^{-1}. \tag{4.1}$$

It suffices to observe that every term on the right hand side (RHS) is nonnegative. □

**Corollary 8.** *The* GRF *prediction operator* $L_{\boldsymbol{u}}^{-1}L_{ul}$ *maps* $\boldsymbol{y_\ell} \in [0,1]^{|\boldsymbol{\ell}|}$ *to* $\hat{\boldsymbol{y}}_{\boldsymbol{u}} = -L_{\boldsymbol{u}}^{-1}L_{ul}\boldsymbol{y_\ell} \in [0,1]^{|\boldsymbol{u}|}$*. When L is singular, the mapping is onto.*

*Proof.* For $\boldsymbol{y_\ell} = \mathbf{1}$, $(L_{\boldsymbol{u}}, L_{ul}) \cdot \mathbf{1} \geq 0$ and $L_{\boldsymbol{u}}^{-1} \geq 0$ imply $\left(I, L_{\boldsymbol{u}}^{-1} L_{ul}\right) \cdot \mathbf{1} \geq 0$, i.e. $\mathbf{1} \geq -L_{\boldsymbol{u}}^{-1} L_{ul} \mathbf{1} = \hat{\boldsymbol{y}}_{\boldsymbol{u}}$.

As both $L_{\boldsymbol{u}} \geq 0$ and $-L_{ul} \geq 0$, we have $\boldsymbol{y_\ell} \geq 0 \Rightarrow \hat{\boldsymbol{y}}_{\boldsymbol{u}} \geq 0$ and $\boldsymbol{y_\ell} \geq \boldsymbol{y}_{\boldsymbol{\ell}}' \Rightarrow \hat{\boldsymbol{y}}_{\boldsymbol{u}} \geq \hat{\boldsymbol{y}}_{\boldsymbol{u}}'$. $\qquad\square$

**Lemma 9.** *Suppose* $L = \begin{pmatrix} L_{11} & L_{12} \\ L_{21} & L_{22} \end{pmatrix}$. *Then* $L^{-1} - \begin{pmatrix} L_{11}^{-1} & 0 \\ 0 & 0 \end{pmatrix} \geq 0$ *and is positive-semidefinite.*

*Proof.* As $L^{-1} \geq 0$ and is PSD, the RHS below is term-wise nonnegative and the middle term PSD (Appendix A.2): $L^{-1} - \begin{pmatrix} L_{11}^{-1} & 0 \\ 0 & 0 \end{pmatrix} = \begin{pmatrix} L_{11}^{-1}(-L_{12}) \\ I \end{pmatrix} (L_{22} - L_{21} L_{11}^{-1} L_{12})^{-1} \left((-L_{21}) L_{11}^{-1}, I\right)$ $\quad\square$

As a corollary, the **monotonicity in** (3.3) for both $R(\cdot) = R_V(\cdot)$ or $R_\Sigma(\cdot)$ can be shown. $\qquad\square$

Both proofs for **submodularity in** (3.4) and Theorem 3 result from more careful execution of matrix inversions similar to Lemma 9 (detailed in Appendix A.4). We sketch **Theorem 3** for example.

*Proof.* Without loss of generality, let $\boldsymbol{u} = \boldsymbol{v} - \boldsymbol{\ell} = \{1, \ldots, k\}$. By Shur's Lemma (Appendix A.3):

$$L_{(\boldsymbol{v}-\boldsymbol{\ell})} := \begin{pmatrix} A_{\boldsymbol{u}} & b_{\boldsymbol{u}} \\ b_{\boldsymbol{u}}^T & c_{\boldsymbol{u}} \end{pmatrix} \quad \Rightarrow \quad \frac{\text{Cov}(y_i, y_k | \boldsymbol{\ell})}{\text{Var}(y_k | \boldsymbol{\ell})} = \frac{(L_{(\boldsymbol{v}-\boldsymbol{\ell})}^{-1})_{ik}}{(L_{(\boldsymbol{v}-\boldsymbol{\ell})}^{-1})_{kk}} = (A_{\boldsymbol{u}}^{-1}(-b_{\boldsymbol{u}}))_i, \forall i \neq k \quad (4.2)$$

where the LHS is a reparamatrization with $c_{\boldsymbol{u}}$ being a scaler. Lemma 9 shows that $\boldsymbol{u}_1 \supset \boldsymbol{u}_2 \Rightarrow A_{\boldsymbol{u}_1}^{-1} \geq A_{\boldsymbol{u}_2}^{-1}$ at corresponding entries. Also notice that $-b_{\boldsymbol{u}_1} \geq -b_{\boldsymbol{u}_2}$ at corresponding entries and so the RHS of (4.2) is larger with $\boldsymbol{u}_1$. It suffices to draw a similar inequality in the other direction, $\text{Cov}(y_k, y_i | \boldsymbol{\ell})/\text{Var}(y_i | \boldsymbol{\ell})$. $\qquad\square$

## 5 A Toy Example and Some Simulations

### 5.1 Comparing V-Optimality and $\Sigma$-Optimality: Active Node Classification on a Graph

To visualize the intuitions described in Section 3.1, Figure 1 shows the first few nodes selected by different optimality criteria. This graph is constructed by a breadth-first search from a random node in a larger DBLP coauthorship network graph that we will introduce in the next section. On this toy graph, both criteria pick the same center node to query first. However, for the second and third queries, V-optimality weighs the uncertainty of the candidate node more, choosing outliers, whereas $\Sigma$-optimality favors nodes with universal influence over the graph and goes to cluster centers.

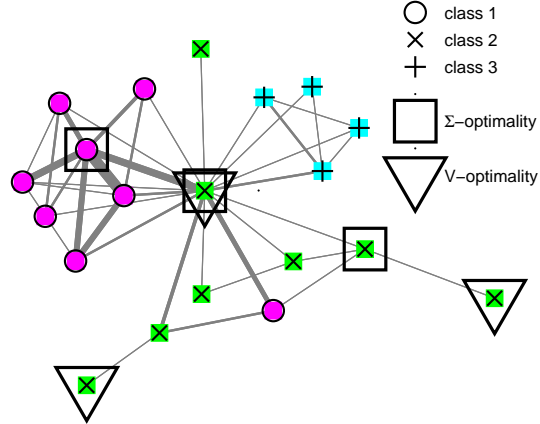

Figure 1: Toy graph demonstrating the behavior of $\Sigma$-optimality vs. V-optimality.

### 5.2 Simulating Labels on a Graph

To further investigate the behavior of $\Sigma$- and $V$-optimality, we conducted experiments on synthetic labels generated on real-world network graphs. The node labels were first simulated using the model in order to compare the active learning criteria directly without raising questions of model fit. We carry out tests on the same graphs with real data in the next section.

We simulated the binary labels with the GRF-sigmoid model and performed active learning with the GRF/LP model for predictions. The parameters in the generation phase were $\beta = 0.01$ and $\delta = 0.05$, which maximizes the average classification accuracy increases from 50 random training nodes to 200 random training nodes using the GRF/LP model for predictions. Figure 2 shows the binary classification accuracy versus the number of queries on both the DBLP coauthorship graph

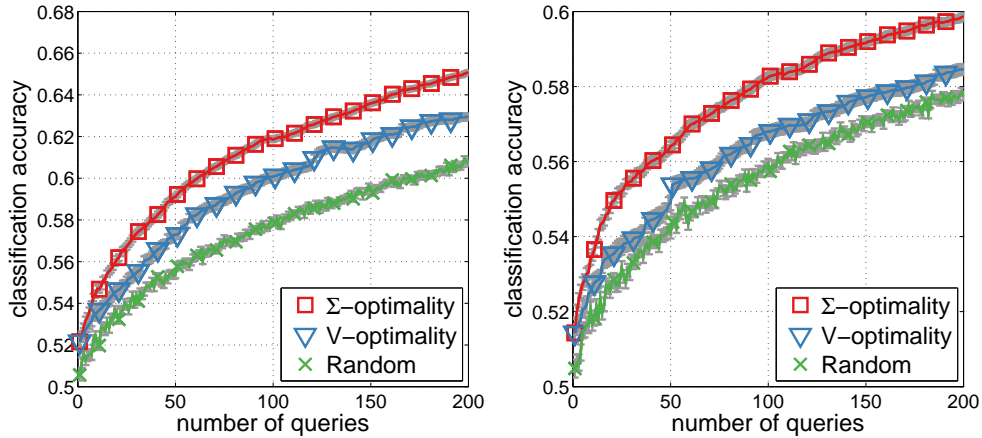

(a) DBLP coauthorship. 68.3% LOO accuracy.     (b) CORA citation. 60.5% LOO accuracy.

Figure 2: Simulating binary labels by the GRF-Sigmoid; learning with the GRF/LP, 480 repetitions.

and the CORA citation graph that we will describe below. The best possible classification results are indicated by the leave-one-out (LOO) accuracies given under each plot.

Figure 2 can be a surprise due to the reasoning behind the $L^2$ surrogate loss, especially when the predictive means are trapped between $[-1, 1]$, but we see here that our reasoning in Sections (3.1 and 5.1) can lead to the greedy survey loss actually making a better active learning objective.

We have also performed experiments with different values of $\beta$ and $\delta$. Despite the fact that larger $\beta$ and $\delta$ increase label independence on the graph structure and undermine the effectiveness of both V/$\Sigma$-optimality heuristics, we have seen that whenever the V-optimality establishes a superiority over random selections, $\Sigma$-optimality yields better performance.

## 6 Real-World Experiments

The active learning heuristics to be compared are:[4]

1. The new **$\Sigma$-optimality** with greedy sequential updates: $\min_{v'} \left( \mathbf{1}^\top (L_{u^k \setminus \{v'\}})^{-1} \mathbf{1} \right)$.
2. Greedy **V-optimality** (Ji & Han, 2012): $\min_{v'} \operatorname{tr} \left( (L_{u^k \setminus \{v'\}})^{-1} \right)$.
3. **Mutual information gain (MIG)** (Krause et al., 2008): $\max_{v'} \left( L_{u^k}^{-1} \right)_{v',v'} \big/ \left( (L_{\ell^k \cup \{v'\}})^{-1} \right)_{v',v'}$
4. **Uncertainty sampling (US)** picking the largest prediction margin: $\max_{v'} \hat{y}_{v'}^{(1)} - \hat{y}_{v'}^{(2)}$.
5. **Expected error reduction (EER)** (Settles, 2010; Zhu et al., 2003). Selected nodes maximize the average prediction confidence in expectation: $\max_{v'} \mathbb{E}_{y_{v'}} \left[ \left( \sum_{u_i \in u} \hat{y}_{u_i}^{(1)} \Big| y_{v'} \right) \Big| y_{\ell^k} \right]$.
6. **Random selection** with 12 repetitions.

Comparisons are made on three real-world network graphs.

1. **DBLP coauthorship network**.[5] The nodes represent scholars and the weighted edges are the number of papers bearing both scholars' names. The largest connected component has 1711 nodes and 2898 edges. The node labels were hand assigned in Ji & Han (2012) to one of the four expertise areas of the scholars: machine learning, data mining, information retrieval, and databases. Each class has around 400 nodes.
2. **Cora citation network**.[6] This is a citation graph of 2708 publications, each of which is classified into one of seven classes: case based, genetic algorithms, neural networks, probabilistic methods, reinforcement learning, rule learning, and theory. The network has 5429 links. We took its largest connected component, with 2485 nodes and 5069 undirected and unweighted edges.

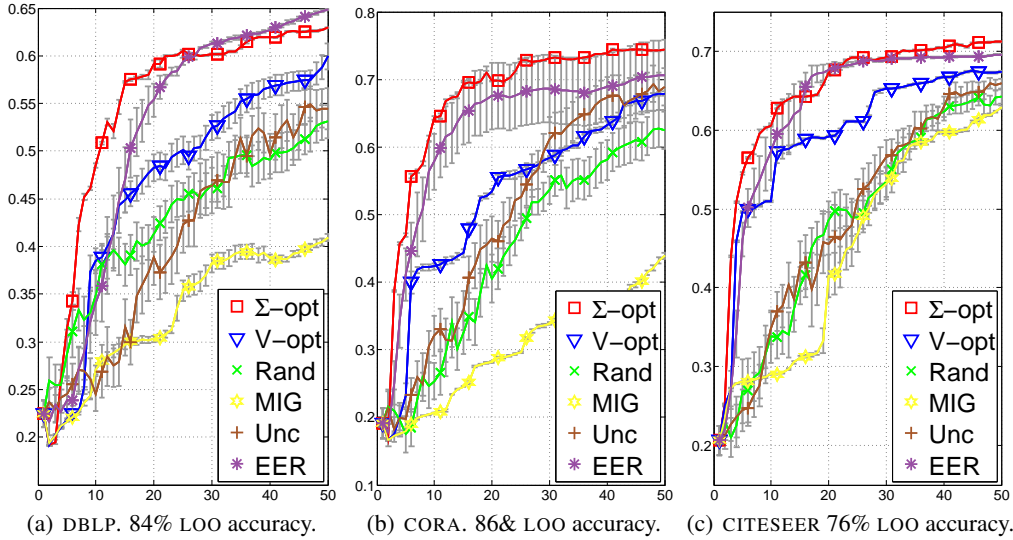

|  |  |  |
|---|---|---|
| (a) DBLP. 84% LOO accuracy. | (b) CORA. 86& LOO accuracy. | (c) CITESEER 76% LOO accuracy. |

Figure 3: Classification accuracy vs the number of queries. $\beta = 1, \delta = 0$. Randomized first query.

3. **CiteSeer citation network**.[6] This is another citation graph of 3312 publications, each of which is classified into one of six classes: agents, artificial intelligence, databases, information retrieval, machine learning, human computer interaction. The network has 4732 links. We took its largest connected component, with 2109 nodes and 3665 undirected and unweighted edges.

On all three datasets, $\Sigma$-optimality outperforms other methods by a large margin especially during the first five to ten queries. The runner-up, EER, catches up to $\Sigma$-optimality in some cases, but EER does not have theoretical guarantees.

The win of $\Sigma$-optimality over V-optimality has been intuitively explained in Section 5.1 as $\Sigma$-optimality having better exploration ability and robustness against outliers. The node choices by both criteria were also visually inspected after embedding the graph to the 2-dimensional space using OpenOrd method developed by Martin et al. (2011). The analysis there was similar to Figure 1.

We also performed real-world experiments on the root-mean-square-error of the class proportion estimations, which is the survey risk that the $\Sigma$-optimality minimizes. $\Sigma$-optimality beats V-optimality. Details were omitted for space concerns.

## 7 Conclusion

For active learning on GRFs, it is common to use variance minimization criteria with greedy one-step lookahead heuristics. V-optimality and $\Sigma$-optimality are two criteria based on statistics of the predictive covariance matrix. They both are also risk minimization criteria: V-optimality minimizes the $L^2$ risk (2.3), whereas $\Sigma$-optimality minimizes the survey risk (2.5).

Active learning with both criteria can be seen as subset optimization problems (2.4), (2.6). Both objective functions are supermodular set functions. Therefore, risk reduction is submodular and the greedy one-step lookahead heuristics can achieve a $(1 - 1/e)$ global optimality ratio. Moreover, we have shown that GRFs serve as a tangible example of the suppressor-free condition.

While the V-optimality on GRFs inherits from label propagation (and random walk with absorptions) and have good empirical performance, it is not directly minimizing the 0/1 classification risk. We found that the $\Sigma$-optimality performs even better. The intuition is described in Section 5.1.

Future work include deeper understanding of the direct motivations behind $\Sigma$-optimality on the GRF classification model and extending the GRF to continuous spaces.

#### Acknowledgments

This work is funded in part by NSF grant IIS0911032 and DARPA grant FA87501220324.

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

## Footnotes

[1] Property p6.4 holds after the first query is done or when the regularizor $\delta > 0$ in (2.1).

[2] Available at http://www.autonlab.org/autonweb/21763.html

[3] In the following, for any vector or matrix $A$, $A \geq 0$ always stands for $A$ being (entry-wise) nonnegative.

[4] Code available at http://www.autonlab.org/autonweb/21763

[5] http://www.informatik.uni-trier.de/~ley/db/

[6] http://www.cs.umd.edu/projects/linqs/projects/lbc/index.html
