[Reviews · NeurIPS 2013]

Submitted by Assigned_Reviewer_5

A new heauristic criterion is provided for active label acquisition for the Gaussian Random Field (GRF) model for semi-supervised classification. The authors show that so-called E-optimal criterion minimizes the survery risk as described by Garnett et al. I very much enjoyed the nice theoretical properties (sub-moudlarity, leading to near optimality guarantees for greedy applications; suppresor free conditions) and insights from analysis (cauchy scwartz inequality based arguments indicate that the criterion is likely to prefer graph cluster centers as opposed to outliers). Experimentally the proposed method outperforms most of the obvious alternatives.

Overall a very nice paper, with sufficient novelty, theoretical rigor, clarity of explanation and reasonable experiments for a good NIPS paper
Summary: A new heauristic criterion is provided for active label acquisition for the Gaussian Random Field (GRF) model for semi-supervised classification. Overall a very nice paper, with sufficient novelty, theoretical rigor, clarity of explanation and reasonable experiments for a good NIPS paper

Submitted by Assigned_Reviewer_6

The authors suggest the use of a criterion, Σ-Optimality, for active
learning in Gauss-Markov random fields. The criterion itself was
originally proposed by Garnett et al for active surveying, but it does
not appear that the submodular property was recognized in that
previous work.

Labeled and unlabeled are embedded in a graph nodes represent both
labeled and unlabled data and edge weights, computed via a kernel,
capture similarity. The motivation for an active approach is that
acquiring labels on the full data set may incur some cost (presumably
greater than computing the edge weights over all data) so a criterion
is used to determine which of the remaining unlabeled data should be
labeled. The authors establish that the criterion satifies the
submodular monotone property and as such greedy selection achieve
(1-1/e) performance relative to optimal selection. The authors should
be careful to note that this optimality is with respect to the
criterion itself and not with respect to classification
accuracy. While the empirical results do give good classification
performance, the criterion itself is only a surrogate.

Several existing criterion are mentioned in the introduction (Settles,
Krause et al, Ji and Han) to which the authors compare their
criterion. One item which is only given limited discusion is the
computational complexity of computing the reward. For example, both
V-optimality and Sigma-optimality require computing the inverse of the
graph Laplacian over the remaining unlabeled data. Some of the other
criterion have lower complexity. This is an important issue since the
problem is cast in terms of costs and since performance of all
criterion will eventually approach each other (since they are
presumably solving the same classification problem following
selection) a fairer comparison would include the cost of computing the
reward. It may be that the proposed method wins here, but it at least
bears some mention.

Why do the authors cite Streeter and Golovin for results regarding
submodular set functions? Shouldn't this be Nemhauser et al?

Section 2.3 misstates several things. The intractability of subset
selection is not a consequence of submodularity, it is simply a
consequence of the combinatorics. Furthermore, this does not mean
that a greedy solution is "required". Other strategies may have lower
complexity and outperform the greedy selection. It is merely, that by
establishing the property, greedy selection has certain guarantees
relative to the optimal selection.

The fact that the criterion of interest was originally proposed by
Garnett et al in 2012 should be mentioned much earlier. This changes
the abstract from "We propos a new criterion" to something more
accurate, such as "We analyze a previously suggested criterion and
demonstrate its utility...".

Establishing the sumodular property as well as the suppressor free
properties are interesting.

Empirical results are suffcient.

Modulo the comments above, the paper is fairly clear. The results are
interesting and the analysis, though limited, represents a
contribution.

minor comments:

Figure 3 lacks a horizontal axis. I assume it is probability of
correct classification, but the authors give an ambiguous description
in the caption.

Summary: The authors establish that a previously proposed criterion is a montonoe submodular function and as such greedy selection achieves performance guarantees relative to optimal selection for an acitve learning problem. Experimental results demonstrate superior performance as compared to previously proposed criterion.

Submitted by Assigned_Reviewer_7

The paper proposes a criterion called \Sigma-optimality for active learning in Gaussian Random Fields (GRFs). V-optimality (proposed before), queries nodes that minimize the L2 loss. The paper argues that for classification problems V-optimality is not ideal (as a surrogate for the 0/1 more appropriate loss).

\Sigma-optimality was proposed before but not for active learning. The paper shows that \Sigma optimality leads to a submodular function and provides a (1 − 1/e) approximation bound to the global optimum. It also shows that experimentally it outperforms V-optimality and other active learning criteria, when a greedy algorithm is used.

Like V-optimality, \Sigma-optimality tend to pick nodes that have high variance or are correlated with nodes with high variance. However, the paper makes the observation based on Eq. 3.7 that \Sigma optimality also prefers nodes that have ‘more influence’, usually nodes in the cluster centers.

Experiments show that on synthetic datasets (generated according to the assumed model) \Sigma-opt outperforms V-opt and random. The graphs are shown for specific model parameters \beta and \delta. It would be interesting to see when \Sigma optimality breaks, in particular when random or V-opt are close (or better) than the proposed approaches. What happens on sparse graphs or highly connected ones?

I am surprised by how badly MIG works and to a lesser extend also for the terrible performance of Uncertainty-based active learning. Is there any explanation for this? It would be useful to include in the paper how MIG was employed and include a discussion to contrast these methods (together with Unc) with the proposed approach.

This paper is clearly written. It is a small extension of previous ideas, in particular the use of V-optimality for the same problem and borrowing the idea of \Sigma-optimality from recent previous work. Its significance is primarily based on the improved performance shown in the experimental evaluation. However, it is not very clear why it outperforms other methods with such ease.
Summary: Overall, the paper is based on using \Sigma optimality as active learning criterion in GRFs (for classification), a well-known problem. \Sigma optimality have been proposed before, thus the mild novelty is in its use for active learning. The proven submodularity guarantee is an incremental contribution.

Submitted by Assigned_Reviewer_8

The paper analyses two optimization objectives for active learning on GRFs - Sigma-Optimality and a V-Objective (that was proposed before, but not analyzed). The authors show that the maximization versions of both the objectives are submodular and hence obtain approximation guarantees on the performance of greedy algorithms. Furthermore, they show that the covariance matrix for GRFs satisfy the Suppressor-free condition.

The paper has some nice theoretical insights, the most interesting of which being that the covariance matrix of GRFs is suppressor-free. The notion of how suppressors play an adverse role in subset selection, and how the absence of suppressors enables good performance guarantees for greedy algorithms has been studied previously, but it was not known whether there was a general class of covariance matrices that satisfied this condition. The authors' result that GRFs do satisfy this condition is quite interesting, both theoretically and practically.

It was not clear how novel the Sigma-Optimality criteria is, given that it seems to have been proposed before in an Active Surveying context. But I liked the fact that the authors analyzed this optimization problem rigorously, and provide approximation guarantees using submodularity.

On the other hand, I think the paper could improve in its writing - there were a few places where it was technically imprecise (though these do not affect correctness of their analysis)
-I found it annoying that the paper was imprecise in distinguishing between minimization and maximization versions of the problem, and kept switching back and forth. The greedy multiplicative performance guarantees and submodularity of the objective are only for the maximization objective, whereas the paper seemed to suggest several times that it worked for the minimization objective (eg. beginning of section 2.3).
-In the first para of page 4, calculating the global optimum is not intractable because of submodularity, and a greedy-solution is not "required".
-In the last para of page 4, the suppressor-free property does not "prove" 2.3 (2.3 is just a definition)
-The proper citation for the greedy bound for submodular-maximization is [Nemhauser et al 1978], not Streeter and Golovin.


Regarding the experiments section, it is very surprising that the Mutual-Information criteria performs even worse than Random.
Summary: The authors analyze two active-learning criteria for GRFs and show performance bounds of greedy algorithms using submodularity of the objectives. They also show that GRFs obey the supressor-free condition. Overall it is a nice paper, and has useful theoretical contributions - though it should fix the imprecise notation/sentences mentioned above.
Author Feedback

Author rebuttal: We thank every reviewer for their thorough understanding of our paper and very helpful comments.

Reviewers 2 and 4 pointed out a few cases where the manuscript makes technically imprecise comments with regards to the properties of submodularity (e.g., that a greedy approach is "required"); we thank the reviewers for their careful notes and will edit the paper accordingly. We will emphasize that the near-optimality we prove is only for the surrogate loss. We will also replace the Streeter/Golovin citation with the more appropriate Nemhauser, et al citation.

We will cite the (Garnett et al 2012) paper earlier and make the link to that paper clearer from the beginning. To summarize, in that work, what we call Sigma optimality (there was no name given previously) was used in the specific context of active surveying. This is natural as it directly minimizes the expected survey loss. The broader application of Sigma optimality we consider was not explored, nor were any of our theoretical contributions previously given.

For the empirical studies, considering the computational costs, all methods require at least one inversion of a submatrix of the graph Laplacian, yet subsequent inversions in S/V-optimality can use rank-one updates as in (2.7). The y-axis in Figure 3 is the percentage of correct classifications.

Our implementation of MIG follows Algorithm 1 of (Krause, et al 2008), using (writing bar(A) for the complement of A) H(y|A) = 1/2 log(var(y|A)) + Const = 1/2 log((Laplacian(bar(A))^-1)_{yy}) + Const and MIG(y|A) = H(y|A) - H(y|bar(A)). We have verified the correctness of our MIG results with independently written code (Andreas Krause's submodular function optimization (SFO) toolbox for MATLAB, with K=inv(Laplacian+10^-6*I)), which produces identical query sequences as our implementation. Notice that in most published successful applications of MIG (e.g., sensor placement), the technique is used with a Gaussian process defined on a low-dimensional (often 2d) Euclidean space. Such setups tend to produce sparse covariance matrices (many entries near 0). In the graph setting, we have a sparse precision matrix, but a dense covariance and typically many more outliers than in these Euclidean examples. Visual inspection of the nodes picked by MIG for our graphs shows that MIG tends to select the central nodes of small and almost-disjoint subgraphs. These nodes both have a very high variance conditioned on the queried nodes and a rather small variance conditioned on the not-yet-queried nodes. What is worse, when these subgraph centers are queried, they neither change our global predictions much nor are they likely to significantly lower the conditional variance of other similar subgraph centers.

The uncertainty sampling results were not too surprising, because uncertainty sampling chooses query nodes close to the decision boundary (which is very large in our graph setting) without regard to the effect of these queries on future predictions. There are typically very many uncertain nodes lying on the periphery of a graph that, after querying, simply do not improve the classifier much. We will add more discussion about the different failure modes for every method we compared against in our experiments.

The limitation of our approach has been briefly explored in our preliminary experiments. Tuning \beta and \delta in the simulations does not change the pattern that Sigma optimality outperforms V optimality (and both outperform random), but only the exact quantitative differences, as the model becomes too hard/easy to learn. Our limitation mainly comes from the basic assumption that the graph forms clusters and that every cluster is mostly one class; we conjecture that our model works best on scale-free graphs. Our preliminary experiments trying to apply Sigma-optimality to Euclidean spaces (based on the Gaussian process and using both the plain sum and the absolute sum of the predictive covariance matrix) did not produce stable results.